# E4BP4 in macrophages induces an anti-inflammatory phenotype that ameliorates the severity of colitis
Yasuko Kajimura[1,7], Akihiko Taguchi ⬚[1,7] ✉, Yuko Nagao[1], Kaoru Yamamoto[1], Konosuke Masuda[1], Kensuke Shibata[2,3,4], Yoichi Asaoka ⬚[5], Makoto Furutani-Seiki[5], Yukio Tanizawa[6] & Yasuharu Ohta[1]

Macrophages are versatile cells of the innate immune system that work by altering their pro- or anti-inflammatory features. Their dysregulation leads to inflammatory disorders such as inflammatory bowel disease. We show that macrophage-specific upregulation of the clock output gene and transcription factor E4BP4 reduces the severity of colitis in mice. RNA-sequencing and single-cell analyses of macrophages revealed that increased expression of E4BP4 leads to an overall increase in expression of anti-inflammatory genes including *Il4ra* with a concomitant reduction in pro-inflammatory gene expression. In contrast, knockout of E4BP4 in macrophages leads to increased proinflammatory gene expression and decreased expression of anti-inflammatory genes. ChIP-seq and ATAC-seq analyses further identified *Il4ra* as a target of E4BP4, which drives anti-inflammatory polarization in macrophages. Together, these results reveal a critical role for E4BP4 in regulating macrophage inflammatory phenotypes and resolving inflammatory bowel diseases.

Macrophages are the central mediators of intestinal immune homeostasis. They regulate inflammation by performing both pro-inflammatory and anti-inflammatory functions[1]. Imbalance between pro- and anti-inflammatory functions leads to prolonged excessive immune activation and intestinal autoimmune diseases, such as inflammatory bowel disease (IBD)[2]. Macrophages are also involved in the interaction between the gut microbiota and IBD[3]. The causal relationship between macrophage malfunction and the resolution of intestinal inflammation, and also the altered monocyte–macrophage differentiation observed in patients with IBD, highlight macrophages as promising therapeutic targets for IBD[4].

A variety of transcription factors have been implicated in the regulation of pro-inflammatory/anti-inflammatory polarization. For example, IRF5 promotes pro-inflammatory polarization[5], while CREB and ELF4 drive the expression of anti-inflammatory-related genes[6,7]. More recently, the transcriptional cofactor YAP has been reported to be involved in the pro-inflammatory/anti-inflammatory switch (i.e., acting on both pro-inflammatory and anti-inflammatory functions) by suppressing the

expression of anti-inflammatory-related genes and increasing the expression of pro-inflammatory-related genes[2]. However, detailed mechanistic insights underlying the regulation of the pro-inflammatory/anti-inflammatory phenotype, and how this is affected in the development of IBD, are still not fully understood.

E4BP4, encoded by *Nfil3*, was initially identified as a transcriptional activator capable of binding to an activating transcription factor (ATF) DNA consensus sequence site[8]. It was subsequently discovered that E4BP4 is a member of the family of clock output genes that generate both behavioral and physiological rhythms[9]. E4BP4 has also been implicated in the development of immune cell lineages such as natural killer cells and T cells[10–12], and we have shown that E4BP4 plays metabolism-regulating roles in pancreatic beta cells and hepatocytes[13–15]. Further, an in silico analysis identified a binding site for E4BP4 in a SNP, rs7234029, which is associated with human IBD[16], and global E4BP4-knockout mice develop colitis with lymphocytes playing a central role[17]. E4BP4 is also induced by the microbial stimulant Lipopolysaccharide in cultured bone marrow-derived myeloid

[1]Division of Endocrinology, Metabolism, Hematological Science and Therapeutics, Department of Bio-Signal Analysis, Yamaguchi University, Graduate School of Medicine, 1-1-1, Minami Kogushi, Ube 755-8505, Japan. [2]Department of Microbiology and Immunology, Yamaguchi University, School of Medicine, 1-1-1, Minami Kogushi, Ube 755-8505, Japan. [3]Department of Molecular Immunology, Research Institute for Microbial Diseases, Osaka University, Suita 565-0871, Japan. [4]Department of Ophthalmology, Graduate School of Medical Sciences, Kyushu University, Fukuoka 812-8582, Japan. [5]Department of Systems Biochemistry in Pathology and Regeneration, Yamaguchi University, School of Medicine, 1-1-1, Minami Kogushi, Ube 755-8505, Japan. [6]Yamaguchi University, 1677-1, Yoshida, Yamaguchi 753-8511, Japan. [7]These authors contributed equally: Yasuko Kajimura, Akihiko Taguchi. ✉e-mail: a.tgc@yamaguchi-u.ac.jp

cells, and E4BP4 suppresses the expression of IL12B, a major pro-inflammatory cytokine[18]. However, the role of E4BP4 in macrophages or as a regulator of intestinal immune homeostasis remains unknown.

In this study, we show that E4BP4-expressing macrophages ameliorate the severity of Lipopolysaccharide-induced colitis, resulting in the sustainment of microbiome diversity with beneficial probiotic bacteria. RNA-sequencing (RNA-seq) and single-cell analysis shows that E4BP4 guides macrophage polarization toward an anti-inflammatory phenotype. We show that E4BP4 binds the *Il4ra* gene, which promotes anti-inflammatory properties, and enhances *Il4ra* expression and chromatin accessibility to ELF4, a transcription factor that suppresses inflammation. These data demonstrate that E4BP4 plays a pivotal role in the phenotypic determination of macrophages and in the severity of intestinal inflammation.

## Results

### Mononuclear phagocyte lineage-specific E4BP4 upregulation reduces the severity of dextran sulfate sodium (DSS)-induced colitis

To examine the role of E4BP4 in macrophages, we generated transgenic mice (M-E4BP4) expressing E4BP4 under the control of the mononuclear phagocyte lineage-specific mouse *Csf1r* promoter (Supplementary Fig. 1a). This approach aims to replicate a more physiological condition, considering previous reports demonstrating the influence of E4BP4 on differentiation in NK cells and its upregulation during inflammation[10,19]. Real-time RT-PCR confirmed a threefold increase in *E4bp4* mRNA in the macrophages isolated from spleen of M-E4BP4 mice compared with those from wild-type (WT) littermates, but not in other cell types (Supplementary Fig. 1b, c).

We used a model of colitis induced by DSS, which induces epithelial damage allowing intestinal bacteria to invade the injured mucosa, leading to sustained mucosal inflammation[20]. This inflammation is characterized by increased macrophage infiltration and excessive production of inflammatory cytokines, enhancing the destructive effect and exacerbating colitis[21]. Although anti-inflammatory macrophages suppress colitis, the main mechanism is not fully elucidated. Nevertheless, reports suggest that alterations in M2 macrophage levels can worsen or improve colitis, indicating that anti-inflammatory macrophages likely ameliorate DSS colitis through complex mechanisms[22,23].

Mice were first administered 2% (weight per volume) DSS dissolved in drinking water for 7 days (acute phase), followed by regular water starting on day 8 until the end of the experiment on day 14 (recovery phase) (Fig. 1a). Substantial weight loss was induced by DSS to a similar degree in both WT and M-E4BP4 mice during the acute phase (Fig. 1b). Following DSS withdrawal, the body weight and disease activity index (DAI) significantly improved in M-E4BP4 mice compared with their WT littermates by day 13 and 14 (Fig. 1b, c), respectively. A shorter colon length is considered a hallmark of experimental colitis[1,20]. Consistent with improved DAI, the M-E4BP4 mice had significantly longer colon lengths compared to WT on day 14 (Fig. 1d, e).

We further evaluated tissue damage in DSS-induced colitis by analyzing tissue sections stained with hematoxylin and eosin (H&E). M-E4BP4 mice had less inflammation and tissue damage than WT at day 14 (Fig. 1f, g). Examination of proliferation markers (Ki67) and apoptosis markers (TUNEL) showed an increase in Ki67 positive cells at 10 days and a decrease in TUNEL-positive cells at 10, 12, and 14 days in M-E4BP4 mice compared to the WT (Fig. 1h, i and Supplementary Fig. 1d). These findings suggest that M-E4BP4 mice have less damage in their intestinal tract and/or may begin to repair it earlier in the recovery phase. Gut microbiota is closely linked to intestinal inflammation, and reduced microbiome diversity has been reported not only in the DSS-induced colitis model but also in human IBD such as Crohn's disease[24,25]. Fecal microbial characterization by 16 S rRNA-amplicon sequencing showed that DSS-induced colitis (day 14) in WT dramatically reduced microbiome diversity as shown previously (Fig. 1j and Supplementary Fig. 1e–g)[24]. Notably, it was observed that the relative abundance of *Lactobacillus* sp. and *Akkermansia* sp. was higher throughout pre- and post-DSS administration (Fig. 1k and Supplementary Fig. 1h).

*Lactobacillus* sp. and *Akkermansia* sp. are reported to play a protective role against DSS-induced colitis, although *Akkermansia* sp. may also exacerbate DSS-induced inflammation and cancer[26,27], see "Discussion". Taken together, these findings suggest that macrophage-specific E4BP4 upregulation may reduce the severity of DSS-induced colitis in association with microbiome preservation and regulation.

### Increased anti-inflammatory gene expression in M-E4BP4 mice colon macrophages

Pro-inflammatory macrophages contribute to disease pathogenesis by secreting pro-inflammatory cytokines and causing tissue damage, whereas anti-inflammatory macrophages protect against excessive inflammation by secreting anti-inflammatory factors[28,29]. We hypothesized that E4BP4 reduced the severity of colitis by modulating pro-inflammatory/anti-inflammatory macrophage polarization.

To test this, we performed single-cell RNA-seq of lamina propria CD45 positive cells isolated from WT and M-E4BP4 mice at day 14 of the recovery phase after DSS-induced colitis (Fig. 2a and Supplementary Fig. 2a, b). Unsupervised clustering was performed on each sample using Surat V4 software, and the cells integrating both WT and M-E4BP4 mice were classified into 10 clusters (Fig. 2b–d)[30]. This approach revealed a decreasing trend in cell number in Cluster 6 in the M-E4BP4 mice compared to WT mice. Since Cluster 6 contains neutrophil-specific genes such as *S100a9*, which plays a role as an inflammatory mediator, this suggests a milder degree of inflammation in the M-E4BP4 mice which is consistent with the DAI and histological scores (Fig. 2d, e and Supplementary Fig. 2c, d). Of the ten clusters, macrophage populations were identified within 2 clusters (1, 3), as they were defined by *Csf1r* expression and reduced expression of other cell type markers (Supplementary Fig. 2c, d). Cluster 1 exhibited a tendency to have a higher cell population in M-E4BP4 mice compared to WT mic and had increased expression of *C1q* and *ApoE*, which are thought to be anti-inflammatory genes (Fig. 2d)[31,32]. In contrast, Cluster 3, which expresses the inflammatory genes *Ifitm3*, *Plac8*, and *Thbs1* (Fig. 2d)[33–35] tended to be lower in M-E4BP4 mice, and surprisingly increased expression of anti-inflammatory genes *Retnla*, *Cd209a*, and *Mgl2* was observed in the inflammatory macrophage population of M-E4BP4 mice in this cluster (Fig. 2e). To next confirm the anti-inflammatory phenotype of the macrophage cluster obtained by the single-cell analysis, we sorted F4/80 CD11b double-positive cells, which are actual macrophages, from the lamina propria (Fig. 2f and Supplementary Fig. 2e, f). We found approximately threefold induction of *E4bp4* mRNA expression by RT-qPCR in colon macrophages isolated from WT mice on day 0 and day 7, suggesting *E4bp4* is induced during inflammation (Fig. 2g). In turn, we performed RNA-seq in colon macrophages isolated from WT and M-E4BP4 mice at the recovery phase (day 14). We identified 4282 differentially expressed genes (Fig. 2h). M-E4BP4 showed increased expression of anti-inflammatory genes such as *Retnla*, *IL10*, and *C1qa*, and *Ccl5*, which is involved in immune escape of cancer, and decreased expression of pro-inflammatory genes such as *Txnrd1*, *Odc1*, *Mmp12*[32,36–40] (Fig. 2h). Gene Ontology (GO) analysis also revealed altered inflammation profiles between WT and M-E4BP4, suggesting an important role for E4BP4 in macrophage-mediated inflammation (Fig. 2i). Together, these results reveal that E4BP4 is upregulated in macrophages during inflammation and may play a role in inducing anti-inflammatory genes.

### E4BP4 induces anti-inflammatory gene expression in macrophages

To corroborate our finding that E4BP4 expression facilitates anti-inflammatory polarization in the transgenic mice model, we next examined whether E4BP4 overexpression by lentiviral vector (E4BP4-TG) and CRISPR/Cas9-mediated E4BP4 knockout (E4BP4-KO) in RAW264.7 murine macrophage cells affect anti-inflammatory polarity formation. E4BP4-TG cells displayed an approximately 20-fold increase in *E4bp4* expression compared to control (CTRL) (Fig. 3a). RNA-seq revealed 708 differentially expressed genes under basal conditions between CTRL and

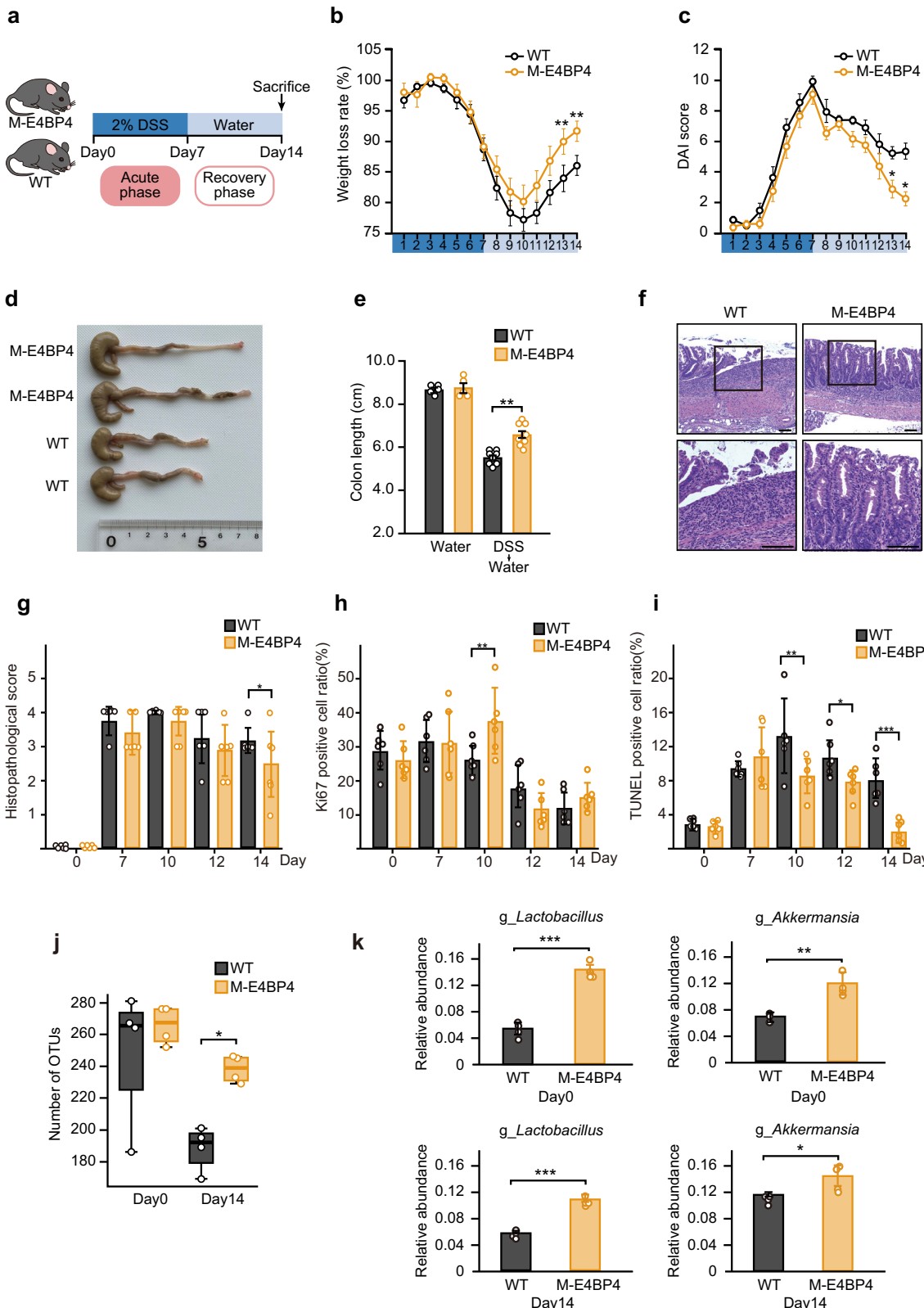

**Fig. 1 | Macrophage-specific E4BP4 upregulation reduces the severity of dextran sulfate sodium (DSS)-induced colitis. a** Experimental timeline for the induction of colitis in mice. Mice were administered 2% (weight/volume) DSS in drinking water for 7 days to induce colitis, followed by regular drinking water for 7 days for recovery. **b**, **c** Daily time courses of body weight change and disease activity index (DAI) ($n = 8$). **d** Representative images of DSS-administered WT and M-E4BP4 mice colons on day 14. **e** Colon length on day 14 with and without 7 days' DSS treatment (water: $n = 5$, DSS + water: $n = 8$). **f** Representative images of the H&E-stained longitudinal colonic sections

on day 14. Scale bars: 100 μm. **g–i** Histological score, Ki67, and TUNEL staining on days 0, 7, 10, 12 and, 14 ($n = 6$). **j** Alpha diversity of stool samples ($n = 4$). The line inside the box represents the median, while the whiskers represent the lowest and highest values within the 1.5 interquartile range (IQR). OTU operational taxonomic unit. **k** The relative abundances of genus *Lactobacillus* and *Akkermansia* in each individual sample obtained from the LEfSe analysis ($n = 4$). All values are expressed as means, error bars reflect SD. Significance was determined by two-way repeated-measures ANOVA, followed by Tukey's post-test ($^*P < 0.05$; $^{**}P < 0.01$; $^{***}P < 0.001$).

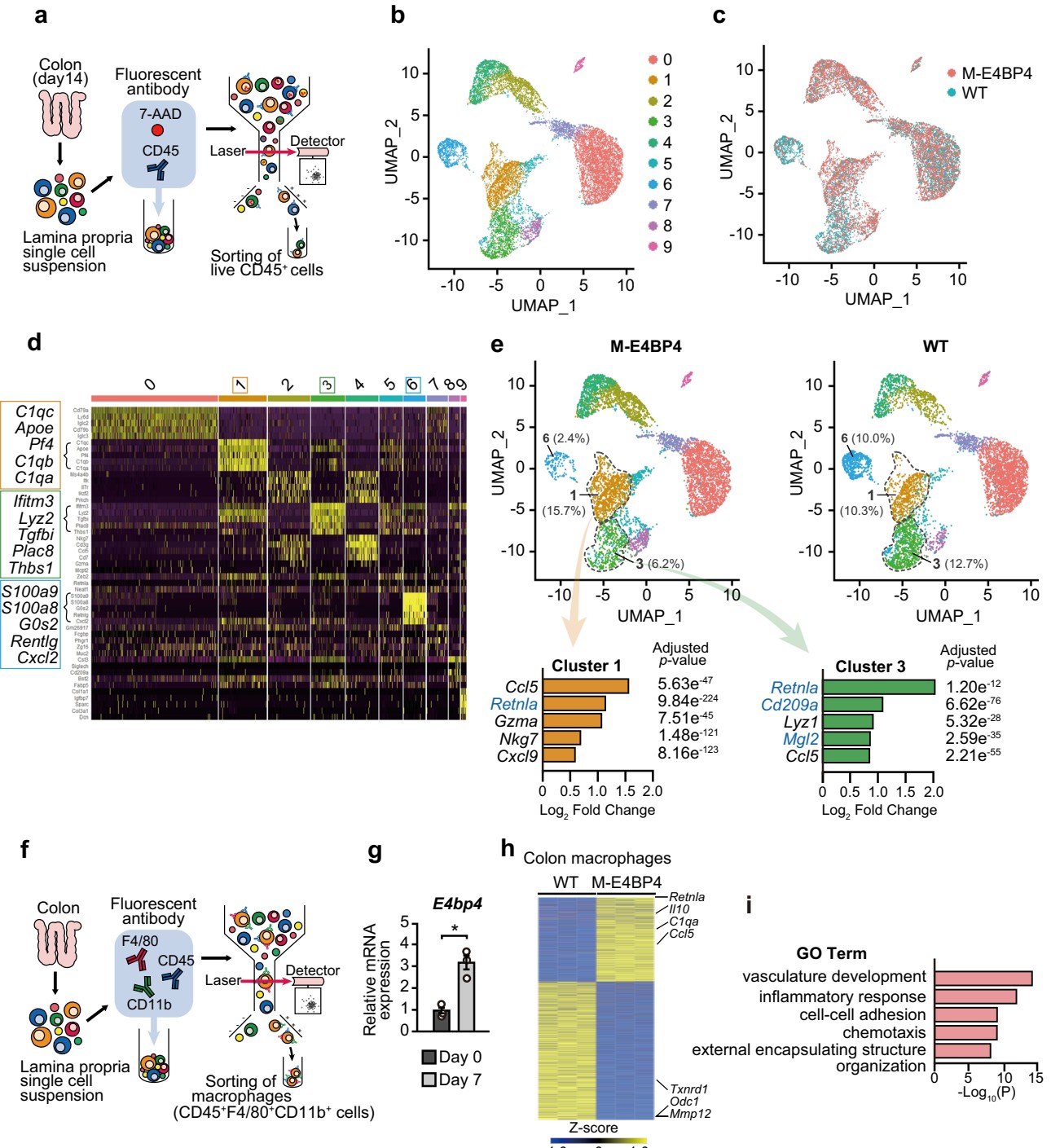

**Fig. 2 | Increased anti-inflammatory gene expression in M-E4BP4 mice colon macrophages. a** Schema for the isolation of CD45 positive cells of the lamina propria by flow cytometry and subsequent single-cell RNA-seq. Colon was dissected on day 14 (recovery phase). **b** A UMAP plot of 13,492 cells from WT (6466 cells) and M-E4BP4 (7,026 cells) colon. Clusters were determined using unsupervised clustering. Each point represents a single cell. **c** A UMAP plot of both WT and M-E4BP4 colon CD45$^+$ cells. Color codes indicate library identity (WT or M-E4BP4). **d** Heatmap of the top five differentially expressed genes among each cluster. Gene groups involved in Cluster 1, 3, and 6 are shown in the expanded view. **e** UMAP plots of WT and M-E4BP4 colon CD45$^+$ cells. Top five differentially upregulated genes in M-E4BP4 vs. WT in Cluster 1 and 3, respectively. Representative M2 macrophage marker genes are shown in blue. The relative frequency of each of Cluster 1 and 3 was

calculated by the number of cells in that cluster compared to the total number of cells and is written in the graph. **f** Schematic overview of macrophage isolation by flow cytometry and subsequent RNA-seq (recovery phase). **g** Relative *E4bp4* gene expression in colon macrophages at day 0 and day 7 (acute colitis phase) using quantitative real-time PCR analysis (*n* = 3). **h** Heatmap showing differentially expressed genes in WT or M-E4BP4 mice colon macrophages at day 14 (recovery phase) using RNA-seq (adjusted *P* value < 0.05). Data show three biological replicates. **i** Top 5 significantly enriched GO Biological Processes in obviously differentially expressed genes (absolute value of Log$_2$ fold change >1). Gating strategies are shown in Supplementary Fig. 5a for (**a**), and Supplementary Fig. 5b for (**f**). All values are expressed as means, error bars reflect SD. Significance was determined by Welch's *t* test for (**b**) ($^*P$ < 0.05).

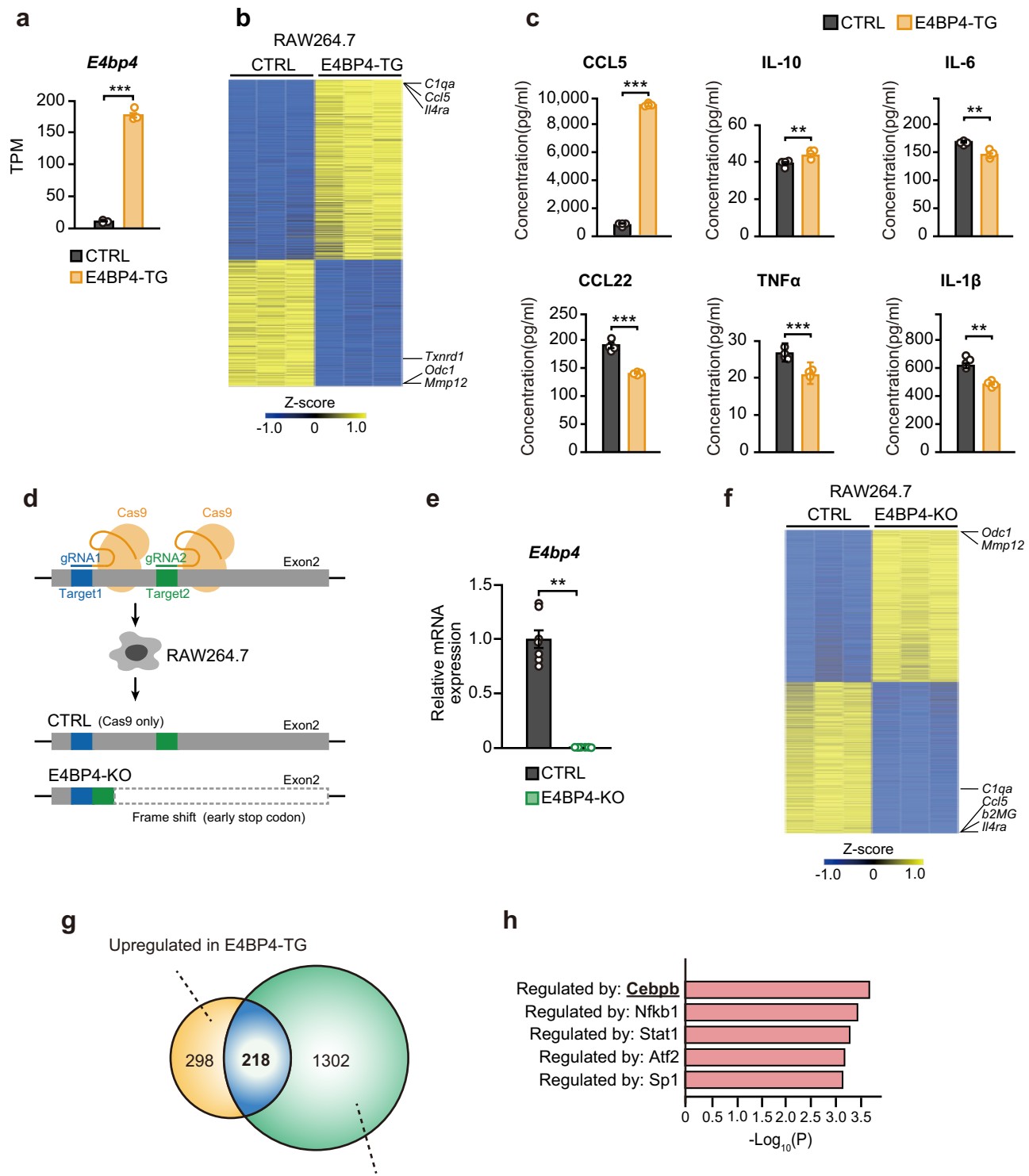

**Fig. 3 | Cell-based studies have revealed that E4BP4 promotes anti-inflammatory genes in macrophages. a** E4bp4 mRNA expression levels in GFP (CTRL) and E4BP4-expressing RAW264.7 cells (E4BP4-TG) ($n = 3$). Expression levels were expressed as TPM (transcripts per million) using RNA-seq. **b** Heatmap showing differentially regulated genes in CTRL and E4BP4-TG RAW264.7 cells using RNA-seq (adjusted $P$ value < 0.05). Data show three biological replicates. **c** Cytokine measurements of cell culture supernatant from CTRL and E4BP4-TG RAW264.7 cells ($n = 4$). **d** Schema of CRISPR/Cas9 E4BP4 (exon 2) ablation in RAW264.7 cells. **e** Quantitative real-time PCR analyses of disrupted E4BP4 using the Q-PCR primer

shown in Supplementary Fig. 3b ($n = 7$). **f** Heatmap showing differentially regulated genes in Cas9 only (CTRL) or E4BP4 knockout (E4BP4-KO) RAW264.7 cells using RNA-seq (adjusted $P$ value < 0.05). Data show three biological replicates. **g** Venn diagram of shared and independent genes whose expression was increased by E4BP4-TG and genes whose expression was decreased by E4BP4-KO.
**h** Transcription factor prediction analysis by TRRUST in 218 genes is shared in (**g**). All values are expressed as means, error bars reflect SD. Significance was determined by Welch's $t$ test ($^{**}P < 0.01$, $^{***}P < 0.001$).

E4BP4-TG RAW264.7 cells, including increased expression of the anti-inflammatory genes *Il4ra* and *C1qa* and immune escape-related *Ccl5* in E4BP4-TG, and decreased expression of pro-inflammatory genes such as *Txnrd*, *Odc1*, and *Mmp12*, consistent in part with the characteristics of colon macrophages (Fig. 3b). Cytokine assays showed elevated levels of CCL5 and IL-10 in M-E4BP4 consistent with RNA-seq results, and decreased secretion of IL-6 and CCL22 (Fig. 3c). Interestingly, a decrease in TNFα and IL-1β levels were observed in E4BP4-TG, whereas expression levels were not significantly different in the RNA-seq analysis, suggesting differential post-transcriptional regulation (Fig. 3c). GO analysis revealed an enrichment of factors in E4BP4-TG that mediate inflammatory responses, suggesting that E4BP4 is involved in macrophage-mediated inflammation (Supplementary Fig. 3a). Next, we established E4BP4-KO RAW264.7 cells and confirmed that E4BP4 expression was absent (Fig. 3d, e and Supplementary Fig. 3b–e). RNA-seq identified 2945 differentially expressed genes between the E4BP4-KO and CTRL RAW264.7 cells under basal conditions, and as expected, genes *Il4ra*, *C1qa*, and *Ccl5* were downregulated and *Odc1* and *Mmp12* were upregulated, the opposite of E4BP4-TG (Fig. 3f). GO analysis also ranked factors mediating the inflammatory response, further supporting a role for E4BP4 in the inflammatory response, as in the E4BP4 overexpression model (Supplementary Fig. 3f). To determine whether these inflammatory changes represent a shift to an anti-inflammatory signature, we performed RNA-seq in RAW264.7 cells stimulated with the M1 polarization compound LPS and cytokine IFNγ, or the M2 polarization cytokine IL-4 (Supplementary Fig. 3g). M2 genes were identified as genes that are upregulated in M2 stimulation and not upregulated in M1 stimulation, with 123 genes registered on the gene set (Supplementary Fig. 3h and Supplementary Table 1). In control RAW 264.7 cells, the M2 gene set was significantly enriched in upregulated genes compared to E4BP4-KO cells (Supplementary Fig. 3i). Furthermore, in E4BP4-TG cells, the M2 gene set was significantly enriched in upregulated genes compared to control RAW 264.7 cells (Supplementary Fig.3j). These results indicate that higher expression levels of E4BP4 are associated with higher expression of the M2 gene set.

We next analyzed the overlapping 218 genes whose expression was both upregulated by E4BP4-TG and downregulated by E4BP4-KO (Fig. 3g). We predicted the transcription factors regulated by these genes using TRRUST as implemented in Metascape and identified Cebpb, a transcription factor downstream of *IL4ra* in macrophages that induces polarization toward anti-inflammation[41–43] (Fig. 3h). These results suggest that CEBPB may be involved in the upregulation of anti-inflammatory genes downstream of *IL4ra* in E4BP4-TG mice.

## Macrophage E4BP4 binds to *Il4ra* and promotes its expression

To further explore the mechanism by which E4BP4 causes macrophage polarization towards an anti-inflammatory phenotype, chromatin immunoprecipitation sequencing (ChIP-seq) and assay for transposase-accessible chromatin using sequencing (ATAC-seq) analyses were performed. First, we performed ChIP-seq on WT RAW264.7 cells with the E4BP4 antibody and identified 56,329 peaks representing potential binding sites of E4BP4 (Fig. 4a). In addition to confirming a clear peak at *Per1*, which E4BP4 has been previously reported to bind[44], E4BP4 binding was enriched in the promoter region and at the established E4BP4 binding motif, indicating the reliability of the ChIP-seq data for E4BP4 (Fig. 4b and Supplementary Fig. 4a, b). GO analysis of these peaks showed an enrichment of genes associated with focal adhesion, which plays a crucial role in macrophage function by facilitating cell adhesion, migration, and interaction with the extracellular matrix (ECM). (Fig. 4c). ATAC-seq was then performed under basal conditions between control and E4BP4-TG RAW264.7 cells to identify regions of chromatin that become more accessible upon overexpression of E4BP4 and may identify upregulated genes. A total of 3802 peaks were detected, with 1.4% of these located within the promoter to TSS region. Several clear ATAC peaks were identified around the *Csf1r* gene, which is abundantly expressed in macrophages (Fig. 4d, e). ATAC peaks are enriched in promoter regions, however peak coverage across all genes showed no difference between CTRL and E4BP4-TG (Supplementary

Fig. 4c). This suggests that E4BP4 may not itself affect chromatin structure, and/or that it may induce either an open or closed chromatin structure depending on the presence of other transcription factors, thereby explaining a potential mechanism of E4BP4 bidirectional expression regulation. Motif analysis showed that the ELF4 binding motif was top-ranked within ATAC peaks in E4BP4-TG compared to CTRL (Fig. 4f). ELF4 is a transcription factor that has been reported to play a protective role against human IBD and has anti-inflammatory properties, which may partly explain the anti-inflammatory effects of E4BP4[7]. Indeed, transcription factor motif analysis using TRRUST on the genes significantly upregulated in the E4BP4-TG revealed the inclusion of Spi1 (PU.1), and Ets2, supporting the notion that the binding motifs of the ETS transcription factor family become open chromatin, potentially contributing to the increased gene expression observed in E4BP4-TG (Supplementary Fig. 4d).

To elucidate a more direct mechanism of action for E4BP4, we extracted genes with increased binding by E4BP4 ChIP-seq, increased expression in E4BP4-TG, and increased peaks in E4BP4-TG by ATAC-seq (Fig. 4g). 27 genes were extracted and subjected to GO analysis, which revealed an enrichment of genes relates to protozoan immunity (Fig. 4h). Four of these genes, including *Il4ra*, are important for anti-inflammatory polarization in macrophages. Thus, we analyzed *Il4ra* further and found multiple ATAC and ChIP peaks around the *Il4ra* gene, demonstrating that E4BP4 binds directly (Fig. 4i). We also confirmed that *Il4ra* expression is downregulated in E4BP4-KO and that this is rescued when E4BP4 expression is returned to WT, together suggesting that *Il4ra* is regulated by E4BP4 (Fig. 4j, k).

Finally, to confirm the beneficial effect of E4BP4 overexpression in colitis, we isolated BMDM (bone marrow-derived macrophages) from WT and M-E4BP4 mice and infused each into WT DSS-induced colitis mice (Fig. 4l). As expected, the severity of colitis was partially reduced in the M-E4BP4 BMDM infused group, confirming the potential therapeutic value of E4BP4 (Fig. 4m and Supplementary Fig. 4e–g).

## Discussion

We have established a role for E4BP4 in colon macrophages by both gain and loss of function studies. We focused on the role of E4BP4 in macrophages as a paradigm of immunoregulation of the colon because macrophages play a critical role in colitis by secreting many cytokines and regulating tissue repair, and because whole body E4BP4 knockout mice develop spontaneous colitis modulating potentially susceptible loci associated with Crohn's disease and ulcerative colitis[16,17].

Macrophages play essential roles during inflammatory conditions, with both pro-inflammatory and anti-inflammatory functions, meaning that their imbalance can result in excessive and persistent inflammation[45,46]. Recently, the resolution of inflammation has been shown to be an active process controlled by the accumulation of anti-inflammatory functions with pro-resolving capacities, signifying an emerging importance of anti-inflammatory macrophages[47,48]. An imbalance between pro- and anti-inflammatory macrophages is related not only to the onset and progression of IBD, but also to many diseases, such as arteriosclerosis, diabetes, and even Alzheimer's disease[49–51]. Thus, our discovery of E4BP4 in modulating macrophage polarization and promoting disease recovery suggests the potential of macrophage-targeted therapy for alleviating inflammatory diseases.

In the present study, in a largely unbiased approach, we found that E4BP4 regulates *Il4ra* expression as a potential mechanism by which E4BP4 induces anti-inflammation in macrophages. This is supported by an independent study showing that administration of BMDM stimulated with the Th2 cytokine IL-4 improved the severity of experimental colitis in mice[52]. IL-4R in macrophages is known to form a complex with IL-4 and induce polarization toward alternatively activated macrophages (AAM, also earlier known as M2)[53]. It has been reported that AAM exerts anti-inflammatory effects and promotes tissue repair and remodeling, angiogenesis, and wound healing involving interaction with fibroblasts[54,55]. In addition, mouse and human AAMs are involved in the induction of regulatory T cells and have been suggested to play an important role in the regulation of adaptive immunity and the establishment of an immune regulatory environment[56,57].

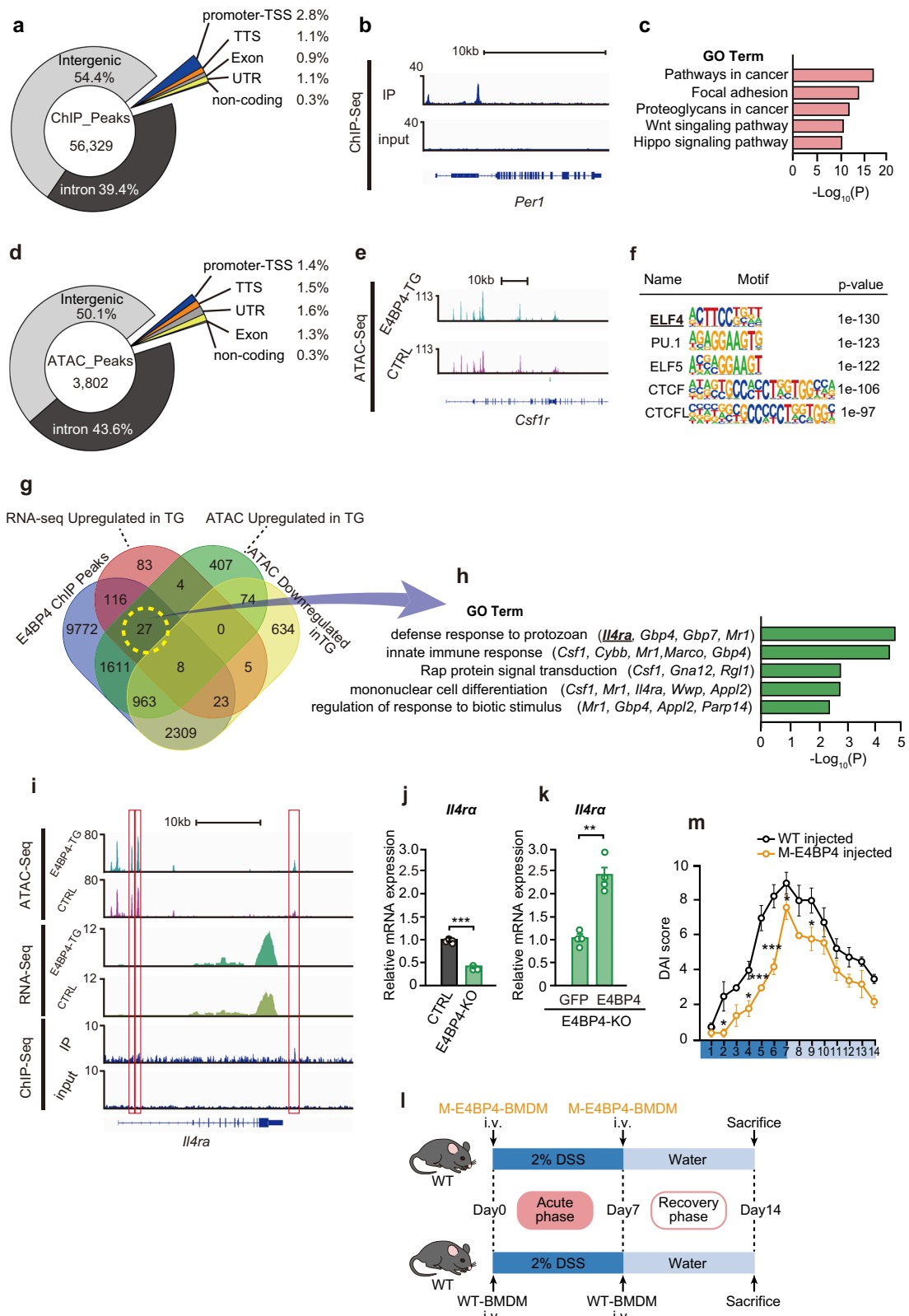

In the current investigation, we focused on both E4BP4 and the gut microbiota, contemplating the possibility that E4BP4 regulates the gut microbial community, particularly contributing to the mitigation of colitis through the increased abundance of Lactobacillus and Akkermansia spp. While there is a wealth of reports highlighting the alleviating effects of *Lactobacillus* spp. on colitis, the role of *Akkermansia* is complex, with reports suggesting both a reduction in colitis and involvement in inflammation and carcinogenesis induced by DSS. Thus, further elucidation is warranted[26,27,58,59].

Furthermore, we confirmed that E4BP4 was induced by DSS administration, and we previously showed that E4BP4 is activated by ER stress in pancreatic beta cells[14]. Furthermore, although E4BP4 is an output clock

**Fig. 4 | Macrophage E4BP4 binds to Il4ra and promotes its expression. a** Genomic distributions of E4BP4 ChIP-seq peaks in WT RAW264.7 cells. **b** IGV browser tracks of E4BP4 ChIP-seq in E4BP4-IP and input DNA (input) along the *per1* gene. Maximum track heights within viewable windows are indicated to the right of each condition. **c** Significantly enriched GO Biological Processes in E4BP4 ChIP-seq peaks. **d** Genomic distributions of ATAC-seq peaks in CTRL and E4BP4-TG RAW264.7 cells along the *Csf1r* gene. **e** Representative IGV browser tracks ATAC-seq in control and E4BP4-TG RAW264.7 cells along the *Csf1r* gene. Maximum track heights within the viewable window are indicated to the right of each condition. **f** Top five motifs enriched within ATAC peaks in E4BP4-TG compared to CTRL. **g** Venn diagram showing overlapping genes with an increased peak at E4BP4-TG by ATAC-seq and E4BP4 ChIP-seq and increased expression by E4BP4-TG by RNA-seq in RAW264.7 cells. **h** Significantly enriched GO Biological Processes in the 27 overlapping genes in (**g**). **i** IGV browser tracks E4BP4 ChIP-seq, ATAC-seq in control and E4BP4-TG RAW264.7 cells along *Il4ra*. Maximum track heights within viewable windows are indicated to the right of each condition. **j** mRNA expression of *Il4ra* in CTRL (Cas9 only) or E4BP4-KO RAW264.7 cells ($n = 4$). Values are expressed as means, error bars reflect SD. Significance was determined by Welch's *t* test ($^{***}P < 0.001$). **k** mRNA expression of *Il4ra* in E4BP4-KO RAW264.7 cells and RAW264.7 cells overexpressing GFP or E4BP4 ($n = 4$). Values are expressed as means, error bars reflect SD. Significance was determined by Welch's *t* test ($^{**}P < 0.01$). **l** Experimental timeline of colitis induction and rescue in mice. Mice were treated with 2% (wt/vol) DSS in drinking water for 7 days to induce colitis and then recovered with normal drinking water for 7 days. Bone marrow-derived macrophage (BMDM) was infused via the tail vein on day 0 and day 7 ($n = 4$ for WT-BMDM, $n = 5$ for M-E4BP4 BMDM). **m** Daily time courses of disease activity index (DAI). Values are expressed as means, error bars reflect SD. Significance was determined by two-way repeated-measures ANOVA, followed by Tukey's post-test ($^{*}P < 0.05$, $^{***}P < 0.001$).

gene, regulation of the D-box, to which E4BP4 binds, has been reported to cause a circadian phase shift[9]. Based on these results, E4BP4 may play a role in adapting the organism to environmental changes by translating environmental information such as circadian cycles, inflammation, and stress.

In the circadian clock field, E4BP4 is conventionally considered to be a transcriptional repressor that reduces *Per2* expression[44]. However, when E4BP4 was first cloned, it was reported to bind the promoter of *Il3* and promote its expression[60]. Interestingly, in the present ATAC-seq studies, although there were variations in expression of genes by RNA-seq between the control and E4BP4-TG cells, no significant differences in chromatin remodeling by ATAC-seq were observed; thus the possibility that E4BP4 acts in a bidirectional manner (i.e., as transcriptional activator and repressor) could not be ruled out (Supplementary Fig. 4c). In the future, the search for co-repressors and co-activators of E4BP4 will be important to determine the precise mechanism of transcriptional regulation by E4BP4.

Limitations of this study include the observed differences in baseline gut microbiota between WT and M-E4BP4 mice and the use of mice expressing E4BP4 in the mononuclear phagocytic lineage, so we cannot exclude the possibility that factors other than macrophages may have influenced the phenotype of the M-E4BP4 mice. However, the results of the rescue experiments using E4BP4-expressing macrophages demonstrated a reduction in the severity of colitis, suggesting that E4BP4 in macrophages does indeed impact the severity of DSS-induced colitis.

E4BP4 is unique in that it not only drives diurnal rhythms of metabolism, but also enables the resolution of inflammation. Further elucidation of the role of E4BP4 in circadian rhythms and macrophages should produce a better understanding of the tissue- and disease-specific roles of E4BP4 and aid in the development of therapeutic strategies against inflammatory diseases such as IBD.

## Methods
### Animals
All mouse experimental protocols were approved by the Ethics of Animal Experimentation Committee at Yamaguchi University School of Medicine. WT C57BL/6 mice were purchased from KYUDO Company (Saga, Japan). Mice were housed in a temperature-controlled ($22 \pm 1$ °C) room under a 12-h light: 12-h dark cycle (LD12: 12). All experiments except for the southern blot analysis were performed using age-matched male littermates. Male mice were used because of previous reports that estradiol prevents DSS-induced colitis[61]. Given that the estrous cycle of female mice is 4 days, this would complicate interpretation of the results from the present experiments.

### Generation of mouse *Csf1r* gene promoter -E4BP4- transgenic mice (M-E4BP4 mice)
The macrophage-specific E4BP4 transgenic mice (M-E4BP4 mice) were generated as described below[62]. The mouse *Csf1r* gene promoter-EGFP-transgenic construct was kindly provided by Dr. David A Hume, University of Edinburgh. The mouse *Csf1r* gene promoter-E4BP4-transgenic construct was created by replacing EGFP-cDNA with mouse E4BP4 (mE4BP4)-

cDNA (1744 bp; ATCC) at the XhoI site. The 12.2-kb mouse *Csf1r* gene promoter-mE4BP4-human growth hormone (hGH) fragment was isolated from the vector by digestion of the plasmid construct with SfiI and HindIII. Microinjection of the purified transgenic DNA into the pronuclei of C57BL/6 mice was performed by UNITECH Company (Chiba, Japan). Founder (G0) mice were identified by PCR and Southern blotting analyses. Genotyping was determined by PCR analysis of genomic DNA extracted from mouse tail using sodium hydroxide and proteinase K (QIAGEN #19133). PCR was performed for 30 cycles at 98 °C for 5 s, at 62 °C for 5 s, and at 72 °C for 10 s with SapphireAmp Fast PCR Master Mix (Takara Bio #RR350A).

### Cell lines
The RAW264.7 cells, a murine macrophage cell line, and the HEK 293TN cells were purchased from KAC Co., Ltd. (Kyoto, Japan) and System Biosciences, respectively. Both cell lines were cultured as a monolayer in 25 mM glucose Dulbecco's modified Eagle's medium (Sigma-Aldrich #D5796) supplemented with 10% fetal bovine serum (FBS, Sigma-Aldrich #172012) and 1% penicillin–streptomycin (FUJIFILM Wako #168-23191) at 37 °C under 5% $CO_2$. The RAW264.7 cells used in experiments were at fewer than 15 passages. Mycoplasma contamination was tested.

### Study design and DSS-induced colitis
8-week-old male mice weighing 23–25 g were used for experimental colitis models. Acute colitis was induced by adding Dextran Sodium Sulfate (MW 36,000–50,000, MP Biomedicals #160110) to the drinking water for 7 days, followed by a 7-day recovery period with the drinking water without DSS and then sacrificed on day14. DSS water was prepared to a concentration of 2.0% in filtered drinking water and changed every 2 days. Control group was provided drinking water from the same source.

### Dispersion of spleen and sorting of hematological cells
Single cell suspensions from spleen were prepared using a gentleMACS Octo Dissociator with Heaters and Spleen Dissociation Kit (Miltenyi Biotec #130-095-926) according to the manufacturer's instructions. Briefly, the spleen of 8–10 weeks old mice was removed, and treated enzymatically and mechanically using the Spleen Dissociation Kit. The cell suspension was washed through 30-μm strainers with phosphate-buffered saline (PBS) containing 0.5% bovine serum albumin (BSA, Sigma-Aldrich #A4503) followed by centrifugation at $300 \times g$ for 10 min at 4 °C and then discarding the supernatants. The spleen cells were resuspended in PBS containing 2% BSA for flow cytometry after removal of red blood cells with the Red Blood Cell Lysis Solution (Miltenyi Biotec #130-094-183). Cells were stained with 7-AAD (BD Biosciences #559925, 1:200), CD3-PE (BioLegend #100203, 1:40), CD45R/B220-APC (BioLegend #103211 1:200), Ly6g/c-FITC (BioLegend #108405 1:100), F4/80-PE (BioLegend #123109, 1:100), CD49b-FITC (BioLegend #108905, 1:100), and CD11c-APC (BioLegend #117309, 1:100) antibodies for 20 min on ice, and then sorted using a FACSAria III Cell Sorter (BD Biosciences) into TRIzol LS Reagent (Thermo Fisher Scientific #10296010) for further RNA extraction. CD3, CD45R/B220, Ly6g/c,

F4/80, CD49b, and CD11c antibodies were used to confirm the enrichment of T cells, B cells, Neutrophils, Macrophages, NK cells, and Dendric cells, respectively. Spleen cells (total spleen) before sorting were used as controls.

## Assessment of colitis

Body weights and feces were evaluated on a daily basis during the DSS-induced acute colitis phase and subsequent recovery phase. Disease activity index was calculated by combining the following scores: weight loss percentage relative to day 0 (pre-DSS treatment) (0: <1%, 1: 1–10%, 2: 11–15%, 3: 15–20%, 4: >20%), stool consistency (0: normal, 2: loose stools, 4: watery diarrhea) and fecal bleeding (0: no blood, 2: slightly bleeding, 4: gross bleeding)[63]. Mice were sacrificed on day 14, and the colon was excised between the ileocecal junction and the anus at the distal end of the rectum. The colon was then separated from the cecum and the colorectal length was measured. Fecal samples were collected on day 0 (pre-treatment) and day 14 (recovery phase of colitis). For histological analyses, mice were sacrificed on days 0, 7, 10, 12, and 14, the total colon was fixed in 4% paraformaldehyde, and longitudinal sections were prepared. H&E, Ki67, and TUNEL staining of sample sections was performed by Morphotechnology Co., Ltd (Hokkaido, Japan). Tissue sections were observed using a KEYENCE BZ-700/BZ-X710 microscope. Disease severity was determined on the basis of a histopathological score using previously described protocols[64]. Colon sections were scored 0-4: 0, normal tissues; 1, mild inflammation in the mucosa with some infiltrating mononuclear cells; 2, increased level of inflammation in the mucosa with more infiltrating cells, damaged crypt glands, and epithelium, mucin depletion from goblet cells; 3, extensive infiltrating cells in the mucosa and submucosa area, crypt abscesses present with increased mucin depletion and epithelial cell disruption; and 4, massive infiltrating cells in the tissue, complete loss of crypts. Ki67 and TUNEL-positive cells were measured using a KEYENCE BZ-X Analyzer 1.4.0.1. Three different sections from the colon tissue sample in each mouse were examined.

## Isolation of murine colonic macrophages

Single cell suspensions from colonic lamina propria were prepared using a gentleMACS Octo Dissociator with Heaters and Lamina Propria Dissociation Kit (Miltenyi Biotec #130-097-410) according to the manufacturer's instructions. Briefly, the colons of 8–10 weeks old mice were removed and cleared of feces, cut longitudinally, and laterally into pieces of 0.5 cm. The colon pieces were incubated in Hank's balanced salt solution without $Ca^{2+}$ and $Mg^{2+}$ (FUJUFILM Wako #082-09865) containing 10 mM HEPES (Nacalai tesque #17557-94), 5 mM EDTA (Wako #345-01865), 1% BSA (Sigma-Aldrich #A4503), and 1 mM dithiothreitol (Sigma-Aldrich #D9779) at 37 °C for 20 min. After vortexing and straining repeatedly in order to remove supernatants containing intraepithelial lymphocytes, the lamina propria pieces were treated enzymatically and mechanically by using the Lamina Propria Dissociation Kit. The cell suspension was washed through 100 μm strainers with PBS containing 0.5% BSA, followed by centrifugation at 300×g for 10 min at 4 °C and then discarding the supernatants completely. The lamina propria cells were resuspended in PBS containing 2% BSA for flow cytometry. Cells were stained with 7-AAD (BD Biosciences #559925, 1:100), CD45-APC (BioLegend #103111, 1:100), CD11b-BB515 (BD Biosciences #564455, 1:100) and F4/80-PE (BioLegend #123109,1:100) antibodies for 20 min on ice, gated for the CD45 + F4/80 + CD11b+ cell population as colonic macrophages, and then sorted using a FACSAria III Cell Sorter (BD Biosciences) into TRIzol LS Reagent (Thermo Fisher Scientific #10296010) for further RNA extraction.

## Isolation and culture of murine bone marrow-derived macrophages (BMDMs)

In brief, 6-8 weeks old male mice were sacrificed, and bone marrow cells were harvested from femurs and tibias. The cells were incubated in RPMI1640-GlutaMAX (Thermo Fisher Scientific #72400-047) supplemented with 10% FBS (Sigma-Aldrich #172012),1% penicillin–streptomycin (FUJUFILM Wako #168-23191), 1% monothioglycerol (FUJUFILM Wako #195-15791)

and 20 ng/ml recombinant murine macrophage colony-stimulating factor (PeproTech #AF-315-02) for 7 days at 37 °C under 5% $CO_2$. After 7 days, BMDMs were harvested.

## Establishment of E4BP4 overexpressing RAW 264.7 cells (E4BP4-TG RAW264.7)

Lentivirus expressing GFP (CTRL) or E4BP4 (E4BP4-TG) were produced by VectorBuilder (Kanagawa, Japan). RAW264.7 cells were plated at a density of $1 \times 10^5$ cells/well (non-treated 24-well plate). Twenty-four hours after incubation, RAW264.7 cells were infected with lentiviruses expressing GFP or E4BP4 at 100 MOI. After infection, the cells were refreshed with fresh medium, and 48 h later, the cells were collected and used for RNA-seq and ATAC-seq.

## CRISPR-mediated E4bp4 deletion in RAW264.7 cells (E4BP4-KO RAW264.7)

Exon 2 of *E4bp4* gene was deleted in RAW264.7 cells by the CRISPR-Cas9 system. All gRNAs were designed using the CHOPCHOP web tool (https://chopchop.cbu.uib.no/). Our CRISPR plasmid vector was constructed by inserting two types of designed gRNA (gRNA-1: 5′-CTGGTAGGATC-TAGGGGTGC-3′, gRNA-2: 5′-CTCGGCGTGTCGGAGAAAAC-3′), Cas9 and puromycin resistance sequences into Lentivirus dual-gRNA vector by VectorBuilder (Kanagawa, Japan). The control plasmid encoded the Cas9 nuclease without the gRNA sequence. Plasmid vectors were transformed into Stellar Competent Cells (Takara Bio #636766) and then purified using the QIAfilter Plasmid Midi Kit (QIAGEN #12243). CRISPR plasmids were packaged in HEK 293TN cells via transfection with the envelope vector pCMV-VSV-G (a gift from Bob Weinberg via Addgene plasmid # 8454) and packaging vector pCMV delta R8.2 (a gift from Didier Trono via Addgene #12263) using Lipofectamine LTX and Plus Reagent (Thermo Fisher Scientific #15338-100). Viral supernatant was then harvested and concentrated using the Lenti-X Concentrator (Takara Bio). Lentiviral titer was measured with the Lenti-X qRT-PCR Titration Kit (Takara Bio #631231). Lentiviral transduction in RAW264.7 cells was performed with TransDux MAX Lentivirus Transduction Reagent (SBI #LV860A-1) at 200 MOI. Puromycin-resistant cells were selected, and then single cell cloning seeded into 96-well plate (non-treated) was performed using a FACSAria III Cell Sorter (BD Biosciences). Genotype was determined by PCR analysis of DNA extracted from clones grown in a 96-well plate. Stable cell lines were assayed for loss of *E4bp4* mRNA and protein.

## Polarization of RAW264.7 cells into M1 or M2 phenotype

RAW264.7 cells were plated at a density of $1 \times 10^5$ cells/well (non-treated 24-well plate). 24 h after incubation, RAW264.7 cells were polarized toward the M1 phenotype with 100 ng/ml LPS (Sigma-Aldrich) and 50 ng/ml IFN-γ (PeproTech), or toward the M2 phenotype with 10 ng/ml IL-4 (PeproTech), or received media alone as the M0 condition for 4 h under normal culture conditions. 4 h after stimulation, cells were washed with PBS and harvested for total RNA extraction.

## Western blotting

The nuclear proteins from RAW264.7 cells were extracted using NE-PER Nuclear and Cytoplasmic Extraction Reagents (Thermo Fisher Scientific #78833) according to the manufacturer's instructions. Protein concentrations were determined by a Pierce BCA Protein Assay kit (Thermo Fisher Scientific #23325) according to the manufacturer's instructions. In all, 35 μg of nuclear protein samples were separated by SDS-PAGE and transferred to PVDF membranes (GE Healthcare #10600023). The membranes were then incubated with antibodies. Antibodies against E4BP4 and Lamin B were purchased from MBL (#PM097, 1:1000) and Santa Cruz Biotechnology (#sc-6217, 1:1000), respectively. Protein signals were visualized using horseradish peroxidase-conjugated secondary antibodies (Jackson ImmunoResearch Laboratories #711-035-152, #115-035-003) and an enhanced chemiluminescence substrate kit (GE Healthcare #RPN2232).

## Cytokine measurements

The samples for cytokine measurements were obtained from culture medium. RAW264.7 cells were seeded in a 12-well plate at a concentration of $2 \times 10^5$ cells/ml in 1.0 ml of medium per well. After incubating for 42 h, 200 μl of media were collected and centrifuged at 1500 rpm for 10 min at 4 °C. In total, 100 μl of supernatants were collected and stored immediately at −20 °C until analysis. The levels of cytokines and chemokines were determined using a fluorescent bead-based multiplex assay kit performed by GeneticLab CO., Ltd. (Hokkaido, Japan). Briefly, standards and samples were loaded into a 96-well plate, and then antibody-immobilized beads were added. Detection antibodies were added followed by analysis on a Luminex200 (Luminex Corporation). The data were reported as median fluorescent intensities (MFI). A standard curve was generated and then cytokine concentrations were calculated.

## Microbiome analysis

Mouse fecal samples were collected on day 0 (pre-treatment) and day 14 (recovery phase of colitis). Analysis of 16 S ribosomal RNA (16 S rRNA) from mouse fecal samples was performed by Bioengineering Lab, Co., Ltd. (Kanagawa, Japan). Briefly, DNA was extracted from mouse feces, and then a 2-step tailed PCR procedure was performed in order to construct libraries. Sequencing libraries were prepared by amplifying the V3/V4 region of the 16 S rRNA gene. Library pools were sequenced on the MiSeq platform using the $2 \times 300$ bp Illumina MiSeq v3 Reagent Kit. Raw sequence data was treated with a microbiome analysis package QIIME2.

## RNA isolation and real-time RT-PCR

Colon macrophages and spleen cells were added to microfuge tubes containing TRIzol LS and TRI Reagent, respectively (Thermo Fisher Scientific #10296010, #TR118) and frozen at −80 °C. Total RNA was extracted using a Direct-zol RNA Microprep Kit (ZYMO RESEARCH #R2060) according to the manufacturer's instructions. Total RNA extraction from RAW264.7 cells was performed with a Quick-RNA Microprep Kit (ZYMO RESEARCH #R1051). cDNA was synthesized using PrimeScript RT Master Mix (Takara Bio #RR036A) according to the manufacturer's instructions. The cDNA of the colon macrophages was amplified using SsoAdvanced PreAmp Supermix (Bio-Rad #1725160) according to the manufacturer's instructions. Quantitative real-time PCR analysis was performed with PowerUp SYBR Green PCR Master Mix (Applied Biosystems #A25742) on the CFX384 Touch Real-Time PCR Detection System (Bio-Rad). The value of each cDNA was calculated using the ΔCt method and normalized to the value of the housekeeping gene, *Gapdh*. Supplementary Table 2 lists qPCR primers.

## Whole transcriptome analysis with RNA-sequencing (RNA-seq)

Following RNA extraction as described above, adequate RNA quality (RIN > 7) was confirmed with a Bioanalyzer (Agilent). mRNA was isolated and fragmented using a NEBNext Poly (A) mRNA magnet Isolation Module (New England BioLabs #E7490) from 200 ng total RNA of RAW264.7 cells, or 20 ng total RNA of colon macrophages on day 14 (recovery phase of colitis), or 100 ng total RNA of BMDMs. cDNA libraries for Illumina sequencing were prepared using a NEBNext Ultra II RNA Library Prep Kit and NEBNext Multiplex Oligos for Illumina (New England BioLabs #E7770, E7335, E7500) according to the manufacturer's instructions. Briefly, first strand cDNA was synthesized using NEBNext Random Primers and NEBNext First Strand Synthesis Reaction Buffer. Second strand cDNA synthesis was performed with NEBNext Strand Synthesis Enzyme Mix. After adaptor ligation, PCR enrichment was performed to amplify cDNA fragments. Library qualities were assessed on a Bioanalyzer (Agilent). Libraries were sequenced on an Illumina NextSeq 550 instrument using 75 bp single-end reads to a depth of >10 million mapped reads. Raw sequencing reads were aligned to the mm10 reference genome using STAR version 2.7.5a[65]. To generate high-confidence results for downstream analysis, we defined genes as substantially expressed when the mean TPM normalized counts >0.1. Differentially expressed RNAs were identified using DESeq2 version 1.12.4 (false discovery rate (FDR)-adjusted

*P* value < 0.05)[66]. Gene Ontology (GO) pathway enrichment analysis was performed on differential genes by Metascape with a criterion of Minimal overlap ≥3, *P* value cutoff <0.01, and Minimal enrichment=1.5)[67]. Gene set enrichment analysis (GSEA) was performed with the GSEA online tool from the Broad Institute version 4.2.1, and results with FDR *P* value <0.25 were considered significant[68,69]. Transcription factor predictions were performed using TRRUST as implemented in Metascape[41]. Morpheus was used for heatmap visualization by giving the TPM values (https://software. broadinstitute.org/morpheus). The Integrative Genomics Viewer (IGV) 2.7.0, Broad Institute) was used for the visual exploration of RNA-seq data.

## Chromatin immunoprecipitation-sequencing (ChIP-seq)

RAW264.7 cells (54 million) were fixed for 30 min with 2 mM disuccinimidyl glutarate and for 10 min with 1% formaldehyde. Nuclei were then isolated in buffer containing 1% SDS, 10 mM EDTA, 50 mM Tris-HCl pH 8.0, and protease inhibitors and sonicated using a Bioruptor (COSMO BIO) to shear chromatin to 200–1000 bp fragments. Protein-DNA complexes were incubated with two kinds of antibodies against E4BP4 (Santa Cruz Biotechnology #sc-28203 and Proteintech #11773-1-AP) and immunoprecipitated with IgG paramagnetic beads (Invitrogen # 11203D). Eluted chromatin was isolated using MinElute PCR Purification Kit (QIAGEN #28004). DNA libraries were prepared from input and ChIP DNA samples using a NEBNext Ultra II DNA Library Prep Kit for Illumina (New England BioLabs #E7645) according to the manufacturer's instructions. The libraries were quantified by both a Bioanalyzer (Agilent) and NEBNext Library Quant Kit for Illumina (New England BioLabs #E7630). They were sequenced on an Illumina NextSeq 550 instrument using 75 bp single-end reads to a depth of >10 million mapped reads. Raw sequencing reads were aligned to the mm10 reference genome using bowtie2 software with default parameter. Peaks for individual replicates were identified for IP over input using the HOMER findPeaks command with setting -style factor. To identify consensus motifs for E4BP4, we scanned 200-bp windows surrounding transcription factor peaks using "findMotifsGenome.pl" with standard background (random genomic sequences sampled according to GC content of peak sequences). To generate the tag density histograms, tags were quantified using HOMER's "annotatePeaks.pl" command with "-size 1000 -hist 5". Peaks were annotated to nearest genes using "annotate-Peaks.pl" and classified as occurring at the promoter (−1 kb to +100 bp from TSS), intragenic (coding exon, intronic, 3'UTR, 5'UTR, TTS, non-coding exon), or intergenic locations. Gene Ontology analysis of associated genes was performed by specifying "-go". The "makeUCSCfile" command was used to generate IGV browser tracks.

## Single-cell RNA-sequencing

After extraction of the colon on day 14 (recovery phase of colitis), single-cell suspensions from colonic lamina propria were prepared as described above and stained with CD45-APC (BioLegend #103111, 1:100) and 7-AAD (BD Biosciences #5599251, 1:100). Live-CD45⁺ cells in colonic lamina propria were sorted using a FACSAria III Cell Sorter (BD Biosciences) and moved to single-cell RNA library preparation immediately. Approximately 10,000 single cells were loaded into each channel of a Chromium Controller (10x Genomics), and then encapsulated within droplets containing barcoded gel beads and reverse transcription reagents using Chromium Next GEM Single Cell 5' Gel Bead Kit v1.1 (10x Genomics #1000167) according to the manufacturer's instructions. Single-cell RNA libraries were constructed using Chromium Next GEM Single Cell 5' Library Kit v1.1 (10x Genomics) according to the manufacturer's instructions. The libraries were sequenced on an Illumina NovaSeq6000. Raw sequencing data was processed with Cell Ranger pipeline version 5.0.0 (10x Genomics). The processed single-cell RNA-seq data were analyzed using Seurat software version 4.0.

## Assay for transposase-accessible-chromatin sequencing (ATAC-seq)

Thirty thousand RAW264.7 cells each expressing E4BP4 or GFP with lentivirus were used in the experiment. Each cell was subjected to

tagmentation Reaction and Purification using an ATAC-Seq Kit Manual Kit (Active Motif #53150) according to the manufacturer's instructions. Tagged DNA was amplified by 10 cycles of PCR with an index primer unique to each sample. The libraries were quantified by a Bioanalyzer (Agilent). They were sequenced on an Illumina NextSeq 550 instrument using 75 bp pair-end reads to a depth of >10 million mapped reads. Raw sequencing reads were aligned to the mm10 reference genome using bowtie2 software with default parameter. Peaks for individual replicates were identified for E4BP4 over GFP using the HOMER "findPeaks" command with setting -style factor. To identify the consensus motif of the ATAC peak that appears in E4BP4 overexpression, we scanned 200-bp windows using "findMotifs-Genome.pl". Generation of histograms, gene ontology analysis, and generation of IGV browser tracks were performed in the same way as for the ChIP-seq experiments.

## Rescue experiment with BMDMs for DSS-induced colitis

BMDM for infusion was isolated from 6- to 8-week-old WT and M-E4BP4 mice. Nine WT male mice were prepared and divided into WT-BMDM infusion group ($n = 4$) and M-E4BP BMDM infusion group ($n = 5$), and were injected intravenously via the tail vein ($1 \times 10^6$ cells in 100 μL PBS) before (day 0) and after (day 7) DSS administration. Body weights and feces were evaluated on a daily basis during the DSS-induced acute colitis phase and subsequent recovery phase.

## Statistics and reproducibility

All measurements were taken on different samples. Statistical analysis was performed using Welch's $t$ test to compare experimental groups with the control group. A one-way ANOVA was used for comparison between three or more groups to establish whether there is a difference between them. A two-way ANOVA was used for comparison between the means of three or more independent groups that have been "split into two variables. Results were expressed as mean ± SD and $P < 0.05$ were considered to be statistically significant.

## Reporting summary

Further information on research design is available in the Nature Portfolio Reporting Summary linked to this article.

## Data availability

All data in this study are available from the lead contact upon request. Sequencing data have been deposited under GEO dataset accession number GSE183075 (RNA-seq of colon macrophage), GSE183076 (RNA-seq of E4BP4 knockout RAW264.7 cells), GSE233929 (RNA-seq of E4BP4 over-expression RAW264.7 cells), GSE183077 (M1-polarized RAW264.7 cells), GSE183078 (M2-polarized RAW264.7 cells), GSE183079 (ChIP-seq), GSE233928 (ATAC-seq), and GSE183529 (single cell RNA-seq). This study did not generate new code. The source data behind the graphs in the paper can be found in Supplementary Data 1. All the uncropped and unedited blot and gel images in this study are included within the Supplementary Fig. 6.

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

## Acknowledgements

We thank Professor Joseph Bass and Dr. Kathryn M. Ramsey at Northwestern University for helpful discussion. We are grateful for the technical contributions of Dr. R. Tsunedomi and Prof. H. Nagano at Yamaguchi University. We thank Y. Wada for skilled technical assistance and Life Science Editors for editorial assistance. This research was supported by grants, from the Japan Society for the Promotion of Science, 23K06401 (to A.T), 19K09006 (to Y.O.), 21H02683 (to M.F.S.), and 19H03710 (to Y.T.). This work was also supported by research grants from the Ube Industries Foundation (to Y.O. and A.T.), Suzuken Memorial Foundation (to A.T.), Fujii Setsuro Memorial Foundation (to A.T.), and a Grant Takeda Science Foundation (to M.F.S.).

## Author contributions

A.T. designed, performed, and analyzed results for all experiments. Y.K. performed and analyzed for experiments of DSS-Colitis, knock out of RAW264.7 cells, single-cell RNA-seq, and BMDM infusion. Y.N. performed experiments on immunostaining, cytokine assays, and overexpression of RAW264.7 cells. K.Y. generated M-E4BP4 mice. A.T., Y.K., M.SF, Y.O., Y.N, and Y.T. wrote the manuscript. K.M. isolated BMDM and performed relevant experiments. Y.A. performed ATAC-seq. K.S. and M.S.F. supervised flow cytometry and single-cell RNA sequence.

## Competing interests

The authors declare no competing interests.
