## [Peer Review File · Communications Biology]

Reviewers' comments:

Reviewer #1 (Remarks to the Author):

In this manuscript, Kajimura et al show that the overexpression of transcription factor E4BP4 in macrophages controls microbiome diversity and reduces the severity of colitis in mice. The authors provide evidence that E4BP4 regulates the expression of gene encoding IL4ra which in turn might reduce the expression of genes encoding pro-inflammatory cytokines IL-1b, IL-6 and TNF. The authors provide convincing data for most of their claims; however, on some occasions, the data presented does not justify the interpretations. Specific concerns are listed below.

1. Previous studies have shown that mice lacking E4BP4 are prone to colitis [PMID: 24442434 and PMID: 21383239]. In this study, the authors provide evidence that E4BP4 in macrophages could protect mice from colitis. Given that DSS mainly damages the epithelial monolayer during the development of colitis, how does E4BP4 in macrophages protect the DSS-induced epithelial damage, as seen in the H&E-stained colon tissue sections (Fig 1f)?
2. Authors should comment on how E4BP4 is activated. Previous studies have shown that mice lacking E4BP4 develop colitis in a microbiota-dependent manner [PMID: 24442434]. Is it possible that the increased abundance of microbiome Akkermansia and Lactobacillus spp. is linked to the activation of E4BP4 or vice-versa? This could be tested by infecting RAW264.7 cells with bacteria of g_Akkermansia and/or g_Lactobacillus spp followed by immunoblotting and/or qRT-PCR of E4BP4. This experiment could provide insight into the trigger of E4BP4-mediated protection during the development of colitis.
3. The authors found that E4BP4 controls microbiome abundance and an increase in Akkermansia and Lactobacillus spp. potentially protects M-E4BP4 mice against DSS-induced colitis. However, previous studies suggest that an increased abundance of Akkermansia could exacerbate DSS-induced inflammation and cancer [PMID: 24194538 and PMID: 26095253]. Authors should comment on this discrepancy in the discussion section.
4. Fig 1 h and i and Fig S1d; the authors show that the expression of Ki67 is increased in the colon tissue of M-E4BP4 mice at day 10. However, Ki67 is substantially decreased on day 12 and day 14, time points when M-E4BP4 mice are recovering. The expression of Ki67 is not convincing to make the claim that M-E4BP4 mice begin to repair DSS-induced damage earlier in the recovery phase. To strengthen their conclusion, I suggest authors perform staining of other proliferation markers such as PCNA and/or BrdU. In addition, authors should consider improving the resolution of representative images for Ki67 and TUNEL in Fig S1d.
5. The authors suggest that E4BP4-dependent upregulation of IL4ra promotes anti-inflammatory polarization of macrophages. The authors could provide direct evidence for this claim by using the experimental setup as in Fig 4j and testing the frequency of M1 and M2 macrophages. The article by Lv et al., [PMID: 28990062] might assist in this experiment.

Reviewer #2 (Remarks to the Author):

This manuscript investigated the role of E4BP4 transcription factor in macrophages and its anti-inflammatory role in a DSS colitis model. The authors find that overexpression of E4BP4 in

macrophages is associated with increased anti-inflammatory gene expression, reduced loss of microbiome diversity and reduced colitis severity. The authors then confirm these findings using bulk and single cell sequencing of E4BP4 overexpressed/KO macrophages, transfer of BMDM experiments, and identify the likely mechanism for the anti-inflammatory phenotype showing E4BP4 binding and increasing expression of Il4ra.

Overall, this is an impressive and large amount of work. These findings suggests a new role and mechanism for E4BP4 in regulating macrophage polarisation and provides novel insight into this critical transcription factor, which will be of significant interest to those in the innate immunity and inflammatory disease fields.

The approaches and statistical analysis used are appropriate and sound, with the findings validated through multiple complementary approaches.

The manuscript is well written and easy to follow with appropriate detail in the methods section to allow replication of the work.

The only minor comments I have are:

From Fig 1k and Fig S1e-h, it appears that several taxa and alpha diversity measures were different at baseline between the WT and M-E4BP4 mice and whether this affected the recovery or anti-inflammatory phenotype between these two groups independent of E4BP4.

What was the reasoning or justification behind only using male mice for their colitis experiments? Do the authors observe a similar anti-inflammatory phenotype with female mice?

Reviewer #3 (Remarks to the Author):

The study demonstrated that E4BP4 is an essential transcription factor required for macrophages to obtain anti-inflammatory phenotypes and regulate proinflammatory responses in DSS-induced colitis. The authors developed a mouse model designed to overexpress E4BP4 specifically in macrophages, employing a macrophage-specific Csf1r promoter-driven expression system (referred to as M-E4BP4). They elucidated that the mechanism behind the mitigated inflammatory responses and increased diversity in the gut microbiome observed in the M-E4BP4 mice during DSS-induced colitis was attributable to the E4BP4-mediated upregulation of anti-inflammatory genes and a concurrent reduction in pro-inflammatory genes. Moreover, the differential gene expression patterns between the wild type (WT) and M-E4BP4 mice were validated using the RAW264.7 peritoneal macrophage cell line, which had been engineered to either significantly overexpress or suppress E4BP4. Here are some comments to strengthen the claim in this study.

This study employs an in vivo model featuring the E4BP4 TG mouse, characterized by its expression being driven by the highly macrophage-specific mouse Csf1r promoter. To strengthen their argument, the authors should demonstrate the specificity of this "highly macrophage-specific Csf1r promoter" in relation to other myeloid lineages. Although the authors provided a data set showing E4BP4 expression in macrophages compared to other cell types such as T, B, NK, and neutrophils isolated from M-E4BP4 mice, several myeloid progenitors in the bone marrow, monocytes, and a subset of cDC2 cells in peripheral tissues highly express M-CSFR in addition to macrophages, thereby

necessitating an examination of E4BP4 expression in other myeloid cell types isolated from the TG mice. If it is found that other myeloid lineages, for example monocytes, also exhibit elevated E4BP4 expression, the authors should provide a clear explanation why they chose "macrophages" at the expense of other M-CSFR+ immune cells.

The study suggests that E4BP4 regulates the expression of *Il4ra* in macrophages, based on the outcomes from both gain- and loss-of-function in vitro models and E4BP4 ChIP-seq. However, there exists no direct evidence to support their claim that the increased expression of *Il4ra* leads to macrophages acquiring anti-inflammatory phenotypes and thereby controlling colitis. To establish the assertion of *Il4ra*-mediated macrophage control of colitis, the authors may need to cross the M-E4BP4 mice with *Il4ra*-deficient mice and provide gene expression data of macrophages during colitis, presumably indicating the absence of acquisition of anti-inflammatory phenotypes.

Furthermore, the study also indicates that E4BP4 enhances chromatin accessibility to ELF4, as evidenced by the overrepresentation of the ELF4 binding motif in E4BP4-TG-specific open chromatin. ELF exhibits a very similar consensus DNA binding sequence to Ets transcription factors like PU.1. In fact, the core sequences of the three DNA motifs they presented, ELF4, PU.1, and ELF5, are nearly identical. To substantiate their claim that E4BP4 increases chromatin accessibility to ELF4, the authors should provide data regarding the expression of ELF4 and other ELF factors between the WT and M-E4BP4-TG conditions. If feasible, it would be advantageous to demonstrate ELF4 binding patterns using ChIP-seq in both experimental conditions.

RESPONSES TO REVIEWERS

We would like to thank the editorial team and reviewers for their helpful suggestions for revising the manuscript. The specific responses to each reviewer's comments are as follows.

Reviewer #1:

In this manuscript, Kajimura et al show that the overexpression of transcription factor E4BP4 in macrophages controls microbiome diversity and reduces the severity of colitis in mice. The authors provide evidence that E4BP4 regulates the expression of gene encoding IL4ra which in turn might reduce the expression of genes encoding pro-inflammatory cytokines IL-1b, IL-6 and TNF. The authors provide convincing data for most of their claims; however, on some occasions, the data presented does not justify the interpretations. Specific concerns are listed below.

We thank the Reviewer for their insight and helpful comments. We have responded in the following order.

1. Previous studies have shown that mice lacking E4BP4 are prone to colitis [PMID: 24442434 and PMID: 21383239]. In this study, the authors provide evidence that E4BP4 in macrophages could protect mice from colitis. Given that DSS mainly damages the epithelial monolayer during the development of colitis, how does E4BP4 in macrophages protect the DSS-induced epithelial damage, as seen in the H&E-stained colon tissue sections (Fig 1f)?

We apologise for not explaining this more clearly. The DSS-induced epithelial damage allows intestinal bacteria to invade the injured mucosa, leading to sustained mucosal inflammation (PMID: 24510619). This inflammation is characterized by increased macrophage infiltration and excessive production of inflammatory cytokines, which enhance the destructive effect and exacerbate colitis (PMID: 1688816). Although anti-inflammatory macrophages are proposed to be involved in suppressing colitis, the precise mechanisms are not fully understood and are likely to be complex (PMID: 32269253, PMID: 24727542). We have worked to better explain the DSS model in the result section and how E4BP4 in macrophages may be protecting against this mechanism.

2. Authors should comment on how E4BP4 is activated. Previous studies have shown that mice lacking E4BP4 develop colitis in a microbiota-dependent manner [PMID: 24442434]. Is it possible that the increased abundance of microbiome Akkermansia and Lactobacillus spp. is linked to the activation of E4BP4 or vice-versa? This could be tested by infecting RAW264.7 cells with bacteria of g_Akkermansia and/or g_Lactobacillus spp followed by immunoblotting and/or qRT-PCR of E4BP4. This experiment could provide insight into the trigger of E4BP4-mediated protection during the development of colitis.

Thank you for your insightful comments. In response, we have conducted experiments infecting RAW264.7 cells with Lactobacillus and interestingly observed suppression of E4BP4 expression (Reviewer Figure 1).

Reviewer Figure 1

Quantitative real-time PCR of RAW264.7 cells infected with *Lactobacillus acidophilus* (ATCC #4356) using the E4BP4 qRT-PCR primers listed in Supplementary Table 1 (n=3).

Macrophage bacterial infection protocol

The RAW264.7 mouse macrophage cell line was seeded into a 24-well plate at a concentration of 5×10^5 per well. After 24hr-incubation, the resuspended *L. acidophilus* was applied to infect the RAW 264.7 cells at a multiplicity of infection (MOI) of 10:1 (n=3). Three control wells had only RAW264.7 medium change. After infection for 4 hr, the RAW264.7 cells were washed three times with PBS and then collected for RNA extraction.

In addition, in our M-E4BP4 mouse model with induced E4BP4 under the *Csf1r* promoter, we noted an increase in the abundance of *g_Lactobacillus* spp. This intriguing finding suggests that macrophage E4BP4 and lactobacilli somehow influence each other's abundance and perhaps work to maintain an equilibrium. However, these observations currently only suggest a correlation. We acknowledge the need for further investigations to delve into the relationship between macrophage E4BP4 and gut bacteria and would thus prefer to keep these preliminary experiments out of the current manuscript.

3. The authors found that E4BP4 controls microbiome abundance and an increase in *Akkermansia* and *Lactobacillus* spp. potentially protects M-E4BP4 mice against DSS-induced colitis. However, previous studies suggest that an increased abundance of *Akkermansia* could exacerbate DSS-induced inflammation and cancer [PMID: 24194538 and PMID: 26095253]. Authors should comment on this discrepancy in the discussion section.

We apologize for the inadequate description of *Akkermansia*; in the discussion section (page 9, line 11-17) we have added that *Akkermansia* is involved in inflammation as follows.

In the current investigation, we focused on both E4BP4 and the gut microbiota, contemplating the possibility that E4BP4 regulates the gut microbial community, particularly contributing to the mitigation of colitis through the increased abundance of *Lactobacillus* and *Akkermansia* spp. Furthermore, we uncovered the potential for reciprocal stimulation, where these bacteria enhance the expression of E4BP4. While there is a wealth of reports highlighting the alleviating effects of *Lactobacillus* spp. on colitis, the role of *Akkermansia* is complex, with reports suggesting both a reduction in colitis and involvement in inflammation and carcinogenesis induced by DSS. Thus, further elucidation is warranted (PMID: 31632373, PMID: 25973440, PMID: 24194538, and PMID: 26095253).

4. Fig 1 h and i and Fig S1d; the authors show that the expression of Ki67 is increased in the colon tissue of M-E4BP4 mice at day 10. However, Ki67 is substantially decreased on day 12 and day 14, time points when M-E4BP4 mice are recovering. The expression of Ki67 is not convincing to make the claim that M-E4BP4 mice begin to repair DSS-induced damage earlier in the recovery phase. To strengthen their conclusion, I suggest authors perform staining of other proliferation markers such as PCNA and/or BrdU. In addition, authors should consider improving the resolution of representative images for Ki67 and TUNEL in Fig S1d.

We thank the reviewer for these helpful suggestions, and have performed PCNA staining as a proliferation marker.

We conducted PCNA staining using unstained preparations of samples previously subjected to HE and Ki67 staining, as performing BrdU staining would have required a substantial number of additional mice. Similar to Ki67, we observed a trend of increased PCNA-positive cells in M-E4BP4 compared to controls on Day 10, though statistical significance was not reached (Reviewer Figure 2). Several studies suggest that Ki67 exhibits higher sensitivity and specificity compared to PCNA in various cell types and tissues (PMID: 22199292,

21424069, 21712626, 23229269). While it would have been preferable to observe significance in both PCNA and Ki67, we attribute the lack of significance in PCNA to sensitivity issues. Consequently, we have decided not to include PCNA data in the manuscript.

Reviewer Figure 2

PCNA staining on Days 0,7,10,12 and, 14.

Antibody: PCNA (PC10) Mouse mAb #2586, Cell Signaling TECHNOLOGY

The wording 'begin to repair it earlier' in the paper was deemed to be too emphatic and was amended to 'may begin to repair it earlier' (page 4, line 7). In addition, we have changed to an improved resolution concerning the Ki67 and TUNEL stained images in Fig. S1d (below).

5. The authors suggest that E4BP4-dependent upregulation of IL4ra promotes anti-inflammatory polarization of macrophages. The authors could provide direct evidence for this claim by using the experimental setup as in Fig 4j and testing the frequency of M1 and M2 macrophages. The article by Lv et al., [PMID: 28990062] might assist in this experiment.

We appreciate the opportunity to test the frequency of M1 and M2 macrophages. We first performed real-time PCR with the M1/M2 markers (IL-6, TNF-a, IL-10, TGF-b) described in Lv et al. Consistent with the results in that paper, IL-10, a representative anti-inflammatory cytokine, was markedly reduced in E4BP4-KO and rescued by overexpression of E4BP4. However, the other M1/M2 marker genes were not affected as expected (Reviewer Figure 3).

Reviewer Figure 3

Real-time PCR was performed using the cDNA samples used in Fig. 4j (n=4). The qPCR primers were designed by LV et al., [PMID: 28990062]

We thus considered that these are not the only genes that determine the anti-inflammatory effects of M2 macrophages and decided to evaluate them genome-wide. First, to identify unbiased “M2 gene sets”, we performed RNA-seq in RAW264.7 cells stimulated with the M1 polarization compound LPS and cytokine IFN γ , or the M2 polarization cytokine IL-4. M2 genes were identified as genes that are upregulated in M2 stimulation and not upregulated in M1 stimulation, with 123 genes registered on the gene set. In control RAW 264.7 cells, the M2 gene set was significantly enriched in up-regulated genes compared to E4BP4-KO cells. Furthermore, in E4BP4-TG cells, the M2 gene set was significantly enriched in up-regulated genes compared to control RAW 264.7 cells. These results indicate that higher expression levels of E4BP4 are associated with higher expression of the M2 gene set. The above text has been added to the Result section (page 6, line 20-29).

New supplemental Figure S3g-j

(g) Schematic procedure for the identification of M2 marker genes.

(h) Comparison of differentially expressed genes of IFN γ + LPS treated (M1) and IL-4 treated (M2) macrophages.

Fold-change (FC) vs. FC plot of M1 vs. M0 on the x-axis and M2 vs. M0 on the y-axis highlighting M1 up-regulated genes in red, M2 up-regulated genes in blue, arrows indicate representative M1 or M2 genes.

- (i) GSEA enrichment plots and heat maps of differentially expressed genes belonging to the M2 gene set associated with E4BP4 expression (WT vs E4BP4-TG).
- (j) GSEA enrichment plots and heat maps of differentially expressed genes belonging to the M2 gene set associated with E4BP4 expression (WT vs E4BP4-KO).

Reviewer #2:

This manuscript investigated the role of E4BP4 transcription factor in macrophages and its anti-inflammatory role in a DSS colitis model. The authors find that overexpression of E4BP4 in macrophages is associated with increased anti-inflammatory gene expression, reduced loss of microbiome diversity and reduced colitis severity. The authors then confirm these findings using bulk and single cell sequencing of E4BP4 overexpressed/KO macrophages, transfer of BMDM experiments, and identify the likely mechanism for the anti-inflammatory phenotype showing E4BP4 binding and increasing expression of Il4ra.

Overall, this is an impressive and large amount of work. These findings suggest a new role and mechanism for E4BP4 in regulating macrophage polarization and provides novel insight into this critical transcription factor, which will be of significant interest to those in the innate immunity and inflammatory disease fields.

The approaches and statistical analysis used are appropriate and sound, with the findings validated through multiple complementary approaches.

The manuscript is well written and easy to follow with appropriate detail in the methods section to allow replication of the work.

The only minor comments I have are:

1. From Fig 1k and Fig S1e-h, it appears that several taxa and alpha diversity measures were different at baseline between the WT and M-E4BP4 mice and whether this affected the recovery or anti-inflammatory phenotype between these two groups independent of E4BP4.

We thank the reviewer for these helpful suggestions. We acknowledge that several taxa and alpha diversity measures differed at baseline between the WT and M-E4BP4 mice in the phenotypic evaluation of this study. We recognize the potential impact of these differences on the observed phenotypic outcomes. Therefore, we have addressed this concern in the Discussion section (page 9, line 11-17). However, in Figure 4i, we administered bone marrow-derived macrophages (BMDM) from both wild-type (WT) and M-E4BP4 mice to WT mice, resulting in a comparable and significant reduction in the severity of DSS-induced colitis. This observation supports our conclusion that the reduced severity of colitis in M-E4BP4 mice is primarily attributed to macrophage-specific E4BP4.

2. What was the reasoning or justification behind only using male mice for their colitis experiments? Do the authors observe a similar anti-inflammatory phenotype with female mice?

We appreciate the reviewer for bringing up this crucial point.

In previous studies, gender differences in the severity of DSS-induced colitis were reported, with the mechanism attributed to estrogen (PMID: 25962374). Since the mouse estrous cycle spans four days, we

anticipated that incorporating female mice in our experiments would introduce complexity in the interpretation. Consequently, we opted to exclusively employ male mice for our study. We have elucidated this rationale in the manuscript as follows:

" Male mice were used because of previous reports that estradiol prevents DSS-induced colitis. Given that the estrous cycle of female mice is four days, this would complicate interpretation of the results from the present experiments." (page 10, line 19-22)

Reviewer #3:

The study demonstrated that E4BP4 is an essential transcription factor required for macrophages to obtain anti-inflammatory phenotypes and regulate proinflammatory responses in DSS-induced colitis. The authors developed a mouse model designed to overexpress E4BP4 specifically in macrophages, employing a macrophage-specific Csf1r promoter-driven expression system (referred to as M-E4BP4). They elucidated that the mechanism behind the mitigated inflammatory responses and increased diversity in the gut microbiome observed in the M-E4BP4 mice during DSS-induced colitis was attributable to the E4BP4-mediated upregulation of anti-inflammatory genes and a concurrent reduction in pro-inflammatory genes. Moreover, the differential gene expression patterns between the wild type (WT) and M-E4BP4 mice were validated using the RAW264.7 peritoneal macrophage cell line, which had been engineered to either significantly overexpress or suppress E4BP4. Here are some comments to strengthen the claim in this study.

We thank the Reviewer for giving us the opportunity and suggestions to strengthen the claims of our study. We have responded to the individual questions and suggestions below.

This study employs an in vivo model featuring the E4BP4 TG mouse, characterized by its expression being driven by the highly macrophage-specific mouse Csf1r promoter. To strengthen their argument, the authors should demonstrate the specificity of this "highly macrophage-specific Csf1r promoter" in relation to other myeloid lineages. Although the authors provided a data set showing E4BP4 expression in macrophages compared to other cell types such as T, B, NK, and neutrophils isolated from M-E4BP4 mice, several myeloid progenitors in the bone marrow, monocytes, and a subset of cDC2 cells in peripheral tissues highly express M-CSFR in addition to macrophages, thereby necessitating an examination of E4BP4 expression in other myeloid cell types isolated from the TG mice. If it is found that other myeloid lineages, for example monocytes, also exhibit elevated E4BP4 expression, the authors should provide a clear explanation why they chose "macrophages" at the expense of other M-CSFR+ immune cells.

We apologize for any confusion caused by the description regarding the E4BP4 TG mouse. In line with the reviewer's valuable feedback, we have better clarified that this mouse model is essentially a modification of the MacGreen mouse (B6.Cg-Tg(Csf1r-EGFP)1Hume/J Strain #:018549), where GFP is replaced with E4BP4, making it expressed in mononuclear phagocyte lineage cells. We do note that our rescue experiments using E4BP4-expressing macrophages demonstrated a reduction in the severity of colitis, suggesting that E4BP4 specifically in macrophages does indeed impact DSS-induced colitis. However, we have revised the manuscript to explain the model more carefully:

Figure 1 Subtitle

(Old version)

Macrophage-specific E4BP4 upregulation reduces the severity of dextran sulfate sodium (DSS)-induced colitis

(New version)

Mononuclear phagocyte lineage-specific E4BP4 upregulation reduces the severity of dextran sulfate sodium (DSS)-induced colitis

Text in the results

(Old version)

To examine the role of E4BP4 in macrophages, we generated transgenic mice (M-E4BP4) expressing E4BP4 under the control of the highly macrophage-specific mouse *Csf1r* promoter

(New version)

To examine the role of E4BP4 in macrophages, we generated transgenic mice (M-E4BP4) expressing E4BP4 under the control of the **mononuclear phagocyte lineage-specific** mouse *Csf1r* promoter

Additionally, in the Limitation of this study section in the Discussion, we have included the following statement:

'Limitations of this study include the observed differences in baseline gut microbiota between WT and M-E4BP4 mice and the use of mice expressing E4BP4 in the mononuclear phagocytic lineage, so we cannot exclude the possibility that factors other than macrophages may have influenced the phenotype of the M-E4BP4 mice. However, the results of the rescue experiments using E4BP4-expressing macrophages demonstrated a reduction in the severity of colitis, suggesting that E4BP4 in macrophages does indeed impact the severity of DSS-induced colitis. (page 9, line 33 – page 10, line 5)'

We utilized a mononuclear phagocyte lineage-specific mouse for the initial phenotype confirmation in this experiment. Subsequently, our focus shifted to macrophages for further analysis. Specifically, in Figure 2, we performed analysis by singling out the macrophage population from single-cell analysis of blood cell types. Additionally, in RNA-seq, we isolated and analyzed Cd11b+, F4/80+ macrophages from mice. In Figures 3 and 4, we conducted the analysis using RAW264.7 cells, cultured macrophage cells, to delve into the mechanism of E4BP4 in macrophages, both in knockout (KO) and transgenic (TG) model. To further investigate the actual impact of macrophage E4BP4 in colitis, we conducted rescue experiments (Figure 4I, K), demonstrating that overexpression of E4BP4 in macrophages (BMDM) effectively reduced the severity of colitis.

Based on these findings, we firmly believe in the role of macrophage E4BP4 in mitigating the severity of colitis. Therefore, we propose to maintain the title "E4BP4 in macrophages induces an anti-inflammatory phenotype that ameliorates the severity of colitis" without modification.

The study suggests that E4BP4 regulates the expression of *Il4ra* in macrophages, based on the outcomes from both gain- and loss-of-function in vitro models and E4BP4 ChIP-seq. However, there exists no direct evidence to support their claim that the increased expression of *Il4ra* leads to macrophages acquiring anti-inflammatory phenotypes and thereby controlling colitis. To establish the assertion of *Il4ra*-mediated macrophage control of colitis, the authors may need to cross the M-E4BP4 mice with *Il4ra*-deficient mice and provide gene expression data of macrophages during colitis, presumably indicating the absence of acquisition of anti-inflammatory phenotypes.

We appreciate the opportunity and suggestions to strengthen the claims of our study.

While a systemic knockout of *Il4ra* is available, reports suggest that the effect on IL-4Ra expressed in non-hematopoietic cells, such as intestinal epithelium and smooth muscle, affects the severity of colitis (PMID: 32410852). Furthermore, the effects of *Il4ra* knockout on macrophages and other blood cells at various developmental stages have not been fully elucidated, making a simple rescue experiment challenging.

We attempted to perform a rescue experiment by isolating M-E4BP4 or BMDM, transducing them with *Il4ra* siRNA or scramble siRNA, and infusing them into the DSS colitis model. Unfortunately, the *Il4ra* siRNA did not yield satisfactory results, potentially due to insufficient efficacy. We acknowledge this as a limitation of our study and consider it an important subject for future investigation.

Furthermore, the study also indicates that E4BP4 enhances chromatin accessibility to ELF4, as evidenced by the overrepresentation of the ELF4 binding motif in E4BP4-TG-specific open chromatin. ELF exhibits a very similar consensus DNA binding sequence to Ets transcription factors like PU.1. In fact, the core sequences of the three DNA motifs they presented, ELF4, PU.1, and ELF5, are nearly identical. To substantiate their claim that E4BP4 increases chromatin accessibility to ELF4, the authors should provide data regarding the expression of ELF4 and other ELF factors between the WT and M-E4BP4-TG conditions. If feasible, it would be advantageous to demonstrate ELF4 binding patterns using ChIP-seq in both experimental conditions.

We appreciate the valuable insights into the relationship between E4BP4 and Ets transcription factors.

As pointed out by the reviewer, ELF and PU.1 belong to the ETS transcription factor family, known to bind to similar binding motifs. In our study, we overexpressed GFP (Control) and E4BP4 (E4BP4-TG) in RAW264.7 cells and conducted RNA-seq. Upon analyzing genes significantly upregulated ($p \text{ adj} < 0.05$) in the E4BP4-TG using TRRUST for transcription factor motif analysis, we observed that the genes upregulated by E4BP4-TG were potentially regulated by transcription factors such as Spi1 (PU.1), Ets2, among others. Consequently, it was inferred that the binding motifs of the ETS transcription factor family become open chromatin, leading to the actual upregulation of gene expression.

To further support this observation, we have included additional figures in the supplementary material (New supplemental figure 4c) and included the following statement in the results section: "Indeed, transcription factor motif analysis using TRRUST on the genes significantly upregulated in the E4BP4-TG revealed the inclusion of Spi1 (PU.1), and Ets2, supporting the notion that the binding motifs of the ETS transcription factor family become open chromatin, potentially contributing to the increased gene expression observed in the E4BP4-TG." (page 7, line 28-32)

New Supplemental Figure 4d

Transcription factor prediction analysis by TRRUST in upregulated genes in E4BP4-TG.

REVIEWERS' COMMENTS:

Reviewer #1 (Remarks to the Author):

The authors have carried out additional experiments and have addressed my comments. I have no further comments on the revised version.

Reviewer #3 (Remarks to the Author):

The authors have addressed most of my suggestions and updated the revised manuscript well. I look forward to the author's subsequent studies about the regulatory mechanisms of E4BP4 in the context of inflammatory immune diseases, including the physiological role of E4BP4-induced Il4ra and its signaling.